

# Assessment of the quality of ACE-FTS stratospheric ozone data

Patrick E. Sheese[1], Kaley A. Walker[1], Chris D. Boone[2], Adam E. Bourassa[3], Doug A. Degenstein[3], Lucien Froidevaux[4], C. Thomas McElroy[5], Donal Murtagh[6], James M. Russell III[7], and Jiansheng Zou[1]

[1]University of Toronto, Department of Physics, Toronto, Canada
[2]University of Waterloo, Department of Chemistry, Waterloo, Canada
[3]University of Saskatchewan, ISAS, Department of Physics and Engineering, Saskatoon, Canada
[4]Jet Propulsion Laboratory, California Institute of Technology, Pasadena, USA
[5]York University, Department of Earth and Space Science and Engineering, Toronto, Canada
[6]Chalmers University of Technology, Department of Space, Earth and Environment, Gothenburg, Sweden
[7]Hampton University, Center for Atmospheric Sciences, Hampton, USA

*Correspondence to*: Kaley A. Walker (kaley.walker@utoronto.ca)

**Abstract.**

For the past 17 years, the Atmospheric Chemistry Experiment Fourier Transform Spectrometer (ACE-FTS) instrument on the Canadian SCISAT satellite has been measuring profiles of atmospheric ozone. The latest two operational versions of the level
2 ozone data are versions 3.6 and 4.1. This technical note characterizes how both products compare with correlative data from other limb-sounding satellite instruments, namely MAESTRO, MLS, OSIRIS, SABER, and SMR. In general, v3.6, with respect to the other instruments, exhibits a smaller bias (which is on the order of ~3%) in the middle stratosphere than v4.1 (~2-9%), however the bias exhibited in the v4.1 data tends to be more stable, i.e. not changing significantly over time in any altitude region. In the lower stratosphere, v3.6 has a positive bias of about 3-5% that is stable to within $\pm1\%$ dec$^{-1}$, and v4.1
has a bias on the order of -1 to +5% and is also stable to within $\pm1\%$ dec$^{-1}$. In the middle stratosphere, v3.6 has a positive bias of ~3% with a significant negative drift on the order of 0.5-2.5% dec$^{-1}$, and v4.1 has a positive bias of 2-9% that is stable to within $\pm0.5\%$ dec$^{-1}$. However, the v4.1 bias in the middle stratosphere is reduced to 0-5% after being corrected for field-of-view modelling errors. In the upper stratosphere, v3.6 has a positive bias that increases with altitude up to ~16% and a significant negative drift on the order of 2-3% dec$^{-1}$, and v4.1 has a positive bias that increases with altitude up to ~15% and
is stable to within $\pm1\%$ dec$^{-1}$.

## 1 Introduction

It has been well established that prior to the implementation of the Montreal Protocol, global stratospheric ozone ($O_3$) concentrations were declining on the order of approximately 5% dec$^{-1}$ (WMO, 2018). Since 1997, after the implementation of the Montreal Protocol, stratospheric $O_3$ concentrations are no longer declining, and now the question remains, *are $O_3$*
*concentrations recovering*? Multiple recent studies (e.g. Harris et al., 2015; Arosio et al., 2019; Szelag et al., 2020), have shown that merged satellite $O_3$ data sets do exhibit positive stratospheric trends over the past decade or so, however the positive



trends may or may not be considered significant depending on how the uncertainties within the individual data sets are treated (SPARC, 2019). When calculating atmospheric trends, one type of uncertainty that needs to be properly characterized is the stability of systematic errors (drift) in the data. This is especially important when merging $O_3$ data sets in order to produce a long-term data record on the order of decades. The ACE-FTS (Atmospheric Chemistry Experiment – Fourier Transform

Spectrometer) satellite instrument's $O_3$ data set is frequently used to help understand the state of stratospheric ozone. It is important to note that this study in no way tries to answer the question of whether $O_3$ concentrations are recovering or not—it is a technical note assessing the quality of ACE-FTS $O_3$ data in the context of $O_3$ recovery.

As of yet, there have been no published studies focusing on characterizing ACE-FTS $O_3$ drift; however, Hubert et al. (2016) compared 14 different $O_3$ data sets from satellite limb sounders to ground and balloon-based measurements in order to

determine the long-term stability of the satellite instruments. They did not find any significant drift in the version 3.0 ACE-FTS data, although the analysis only included ACE-FTS data from 2004-2010. Rahpoe et al. (2015) calculated relative drifts between six different $O_3$ data sets from satellite limb sounders. Similarly, the ACE-FTS version 3.0 data product used only spanned 2004-2010, and no significant drift was identified, likely due to using too short a time series.

In this study, ACE-FTS $O_3$ profiles have been compared to correlative data sets from satellite-based limb sounders that overlap

in time with essentially the entire ACE-FTS mission, i.e. MAESTRO (Measurement of Aerosol Extinction in the Stratosphere and Troposphere Retrieved by Occultation), Aura MLS (Microwave Limb Sounder), OSIRIS (Optical Spectrograph and Infrared Imaging System), SABER (Sounding of the Atmosphere using Broadband Emission Radiometry), and SMR (Sub-Millimetre Radiometer). These five instruments were all in orbit and measuring ozone in 2004, the year when ACE-FTS began making measurements, and are still in operation.

A description of the instrumentation can be found in Section 2, and the methodology is described in Section 3. The global and regional results for ACE-FTS $O_3$ comparisons are discussed in Section 4, and all the results are summarized in Section 5.

## 2 Instrumentation

Table 1 gives an overview of some of the key details related to the satellite instruments used in this study.

### 2.1 Instruments on SCISAT

The Canadian SCISAT satellite was launched into a non-sun synchronous, high-inclination orbit in 2003 at an altitude of ~650 km. On board are two instruments, ACE-FTS and MAESTRO, both of which use solar occultation viewing geometry to measure profiles of atmospheric state parameters. Both instruments began making regular measurements in February 2004 and are still in operation as of 2021.





### 2.1.1 ACE-FTS

The ACE-FTS instrument (Bernath et al., 2005) is a high-spectral-resolution (0.02 cm$^{-1}$) spectrometer viewing the Earth's limb in the infrared between 750 and 4400 cm$^{-1}$. Since February 2004, ACE-FTS has been providing pressure and temperature profiles and volume mixing ratio (VMR) profiles of over 30 atmospheric trace gases and more than 20 isotopologue species.

ACE-FTS profiles the limb between 5 and 150 km with a vertical sampling of ~2 to 6 km, depending on the orbital geometry, and the vertical extent of the instrument field-of-view at the tangent altitude is on the order of 3-4 km. Two different versions of the level 2 ACE-FTS O$_3$ data are used in this study, version 3.6 (v3.6) and version 4.1 (v4.1).

The retrieval algorithm for trace species concentrations is described by Boone et al. (2005; 2013; 2020) and it uses a non-linear, least-squares, global-fitting technique that fits observed spectra to forward modelled spectra in species dependent

microwindows. The modelled spectra are calculated using spectral line parameters from the HITRAN2004 database (Rothman et al., 2005) with various updates (Boone et al., 2013) for version 3.6, and the v4.1 retrieval uses HITRAN2016 (Gordon et al., 2017). In March 2021, the processing environment for the v4 retrievals was changed, and the current operational version is v4.1/4.2, with no significant differences in the retrieval results between v4.1 and 4.2.

Both versions 3.6 and 4.1 of the ACE-FTS O$_3$ retrieval use 40 microwindows between 829 and 2673 cm$^{-1}$, and account for

CFC-12, HCFC-22, CFC-11, N$_2$O, CH$_4$, HCOOH, and various isotopologues as interfering species. The retrievals have a lower altitude limit of 5 km and an upper altitude limit of 95 km. Horizontal homogeneity is assumed in the retrievals, and diurnal variation along the line of sight is not accounted for. ACE-FTS v2.2 O$_3$ was validated by Dupuy et al. (2009), and Sheese et al. (2017) compared ACE-FTS v3.5 O$_3$ to correlative data from MIPAS and MLS. The ACE-FTS v2.2 O$_3$ profiles are known to have a positive bias on the order of 15-20% in the upper stratosphere – lower mesosphere, near 50-60 km, and on average

typically agree with correlative data sets to within ±5% in the middle lower to middle stratosphere (~20-40 km) with a slight positive bias on the order of a few percent. Compared to Aura MLS and MIPAS, the average bias found for ACE-FTS v3.5 is within 2% from 10-45 km and up to 19% between 46 and 60 km.

### 2.1.2 MAESTRO

The MAESTRO instrument (McElroy et al., 2007) consists of two spectrophotometers designed to cover the spectral range

210-1025 nm, with 1.5-2 nm spectral resolution. The solar occultation measurements are used to retrieve profiles of aerosol extinction and concentrations of O$_3$, NO$_2$, and H$_2$O in the upper troposphere and stratosphere (~5-52 km) with a vertical resolution on the order of 1 km. The O$_3$ retrieval algorithm (McElroy et al., 2007) fits apparent optical depth spectra to modelled spectra in order to derive slant column densities. The forward model assumes temperature independent O$_3$ and NO$_2$ absorption cross-sections from Burrows et al. (1998; 1999). The slant columns are then used in a Chahine inversion technique (Chahine,

1968) to retrieve O$_3$ profiles.

This study uses v3.13 of the MAESTRO O$_3$ data, which has not yet been independently validated. Kar et al. (2007) compared version 1.2 MAESTRO O$_3$ data to correlative data from ozonesondes and satellite instruments, including ACE-FTS. They





found that between 16 and 50 km MAESTRO typically agreed with ozonesonde data to within ~5–10% and that MAESTRO and ACE-FTS tended to agree within ~5–15% in the stratosphere up to 50 km. They also found that while MAESTRO sunset profiles typically agreed with satellite measurements to within 5–10% in the 16–40 km region, MAESTRO sunrise profiles exhibited a positive bias of up to ~20-30% in the 40–55 km region and a negative bias of ~5–15% near 20–30 km. Similar

results were found by Dupuy et al. (2009) when comparing v1.2 MAESTRO $O_3$ data to correlative data from satellite, balloon, airborne, and ground-based instruments. Hubert et al. (2016) found no significant drift between MAESTRO v1.2 and balloon borne measurements for the period 2004-2010, again likely due to the short time period.

## 2.2 MLS on Aura

The Aura satellite was launched in 2004 into a sun-synchronous orbit (ascending node of 13:45 LT) near 700 km. On board is

the MLS instrument (Waters et al., 2006), which observes thermal emission in the Earth's limb in a spectral range of 118 GHz to 2.5 THz in order to retrieve profiles of temperature, geopotential height, and concentrations of over 15 atmospheric trace species on a vertical pressure grid.

The MLS v5 $O_3$ profiles, which are used in this study, are retrieved from observations from the 240-GHz radiometer measurements and are scientifically useful between pressure levels of 261 and 0.001 hPa (~10 and 90 km), with a vertical

resolution of 2.5-4 km in the stratosphere (Livesey et al., 2020). The MLS retrieval algorithm, as described by Livesey et al. (2006; 2020), uses a Newtonian optimal estimation technique (Rodgers, 2008), with a forward model that does not assume horizontal homogeneity, given the Aura MLS line-of-sight viewing conditions (Livesey and Read, 2000). The absorption cross-sections used in the forward model are from the JPL Spectral Line Catalogue (Pickett et al., 1998) with updates.

The v2.2 $O_3$ product has been validated by Jiang et al. (2007), Froidevaux et al. (2008), and Livesey et al. (2008); in the

stratosphere, the results of those validation studies are generally applicable to the v5.1 stratospheric $O_3$ data (Livesey et al., 2020). When comparing to ground-based $O_3$ profiles, Hubert et al. (2016) found the v3.3 MLS $O_3$ data to be stable within 2% dec$^{-1}$ in the upper stratosphere and within 1.5% dec$^{-1}$ in the middle stratosphere.

## 2.3 Instruments on Odin

The Odin satellite (Murtagh et al., 2002) was launched in 2001 into a sun-synchronous orbit (ascending node of 06:00 LT) at

an altitude of ~600 km. There are two limb sounding instruments aboard the satellite that are currently in operation, OSIRIS (Llewellyn et al., 2004) and SMR (Frisk et al., 2003).

### 2.3.1 OSIRIS

The OSIRIS instrument, which uses an optical spectrograph operating in the spectral range of 280-810 nm to observes Rayleigh and Mie scattered sunlight in the Earth's limb and retrieves profiles of $O_3$, $NO_2$, and BrO concentrations and aerosol extinction.

Scans are made between altitudes of approximately 7 and 110 km with a vertical field-of-view of approximately 1 km. As detailed by Bourassa et al. (2012), $O_3$ concentrations are retrieved using the Multiplicative Algebraic Reconstruction



Technique (MART) technique (Roth et al., 2007; Degenstein et al., 2009) between approximately 10 and 60 km, with a vertical resolution on the order of 2 km, and uses pressure and temperature profiles from the European Centre for Medium-Range Weather Forecasts (ECMWF) ERA-Interim reanalysis (Dee et al., 2011). Within the $O_3$ retrieval, UV and visible absorption is taken into account, and aerosols and $NO_2$ are both considered interfering species and are retrieved simultaneously.

The version 5.10 OSIRIS data are used in this study. Hubert et al. (2016) found there to be a significant positive drift in the v5.07 OSIRIS $O_3$ data above 20 km with respect to ozonesonde and lidar data. Between 22 and 35 km, the OSIRIS drift is on the order of 1-3% dec[-1], and above 37 km the positive drift increases to 8% dec[-1] near 42 km. However, Bourassa et al. (2018) determined that these drifts were due to a systematic error in the pointing knowledge and showed that the drifts were significantly reduced in v5.10. Adams et al. (2014) determined that throughout the stratosphere the v5.07 OSIRIS drift relative

to the GOMOS (Global Ozone Monitoring by Occultation of Stars) was less than 3% dec[-1], and v5.07 was shown to be in excellent agreement (within < 5%) with coincident SAGE II profiles throughout the stratosphere by Adams et al. (2013).

### 2.3.2 SMR

The SMR instrument uses four tuneable receivers within 486-581 GHz and a mm-wave receiver at 119 GHz to observe thermal emissions in the Earth's limb. The SMR observations are used to retrieve profiles of temperature and $O_3$, $H_2O$, $N_2O$, $HNO_3$,

and ClO concentrations. Three different $O_3$ products are retrieved, one from emissions measured at the 544.6 GHz line, one from the 501.8 GHz, and one from the 488 GHz line; however, only the 544 GHz $O_3$ retrievals are used in this study. The $O_3$ retrieval algorithm (Urban et al., 2005) makes use of measurements in a 1 GHz band (centred at 544.6 GHz) and uses a Newtonian Levenberg-Marquardt optimal estimation technique (Rodgers, 2008). The forward model used in the retrievals is the open-source ARTS (Atmospheric Radiative Transfer Simulator) forward model (Buehler et al., 2005). Version 3.0 $O_3$, used

in this study, is retrieved at altitudes between 11 and 109 km, with a vertical resolution of ~2-3 km. Jones et al. (2007) compared v2.1 501 GHz $O_3$ to correlative data from satellite and balloon measurements, and Sagi and Murtagh (2016) compared the v2.1 501 and 544 GHz $O_3$ data sets, showing that the two data sets are within 10% of each other at altitudes between 15 and 40 km. Sagi et al. (2017) used v2.1 544 GHz $O_3$ data in their study on $O_3$ depletion in the Northern hemisphere.

### 25   2.4 SABER on TIMED

The TIMED (Thermosphere, Ionosphere, Mesosphere Energetics and Dynamics) satellite was launched in 2001 into a non-sun-synchronous orbit, sweeping through 24 h of local time every 36 days. On board is the SABER instrument (Russell et al., 1999), which uses observations of infrared emissions to retrieve $O_3$ concentrations throughout the stratosphere to lower thermosphere in a 9.6µm channel and in the mesosphere to lower thermosphere in a 1.27µm channel, both with vertical

resolution of ~2 km. This study makes use of the v2.0 $O_3$ 9.6µm data, the retrieval of which is described by Rong et al. (2009) and uses an iterative onion peel retrieval method, taking into account non-LTE (local thermodynamic equilibrium) effects.





The version 1.07 $O_3$ data products are discussed and validated by Rong et al. (2009), where they found the 9.6μm $O_3$ data to have a positive bias in the stratosphere on the order of 5-15% (above what can be explained by systematic uncertainties). Corrections for this bias have been formulated and are under active study by the SABER team. An updated data set is expected to be produced in the next year (Personal communication, James Russell, Hampton University, Hampton, VA). To the authors'

knowledge, no studies have been published on the differences between the v1.07 and v2.0 9.6μm $O_3$ data products, however Fytterer et al. (2015) discuss how v1.07 can be used to supplement v2.0 where there are data gaps without the need for systematic corrections. The SABER $O_3$ data was not considered in the drift analysis study of Hubert et al. (2016).

## 3 Methodology

In this and the following sections, the term INST will be used in general to refer to any of the instruments (other than ACE-

FTS). The coincidence criteria used in all of the comparisons were such that ACE-FTS and INST profiles must have been measured within 6 hours and 300 km of each other. These coincidence criteria were chosen in order to ensure that coincident profiles were as close to common-volume as possible while still having enough profiles to ensure results are statistically significant, and, as discussed by Sheese et al. (2021), these criteria keep the estimated $O_3$ geophysical variability (1σ) between ACE and Odin measurements to less than 5%. The geophysical variability between ACE-FTS and MLS and between ACE-FTS

and SABER is assumed to be similar to the geophysical variability between ACE and Odin.

Prior to analysis, all MAESTRO, OSIRIS, SABER, and SMR profiles, which do not have simultaneously retrieved pressure values, have been linearly interpolated onto the ACE-FTS 1-km grid. The MLS profiles have been interpolated to corresponding coincident ACE-FTS pressure values (on a 1-km grid) in order to avoid any uncertainties inherent in the retrieved MLS geopotential height values. None of the profiles were vertically smoothed, as the vertical resolutions of all the

instruments are relatively similar and smoothing the data has little to no effect on comparison results (e.g. Sheese et al., 2016). In cases where an ACE-FTS profile was coincident with multiple profiles from an INST data set, only the profile measured closest in latitude to the ACE-FTS occultation was used.

For direct comparisons between ACE-FTS and INST, the relative mean differences are calculated with respect to ACE-FTS data as

$$diff = 2\frac{\sum_i^n X_i - Y_i}{\sum_i^n X_i + Y_i}, \tag{1}$$

where $X_i$ are the ACE-FTS values at a given height and $Y_i$ are the corresponding INST values. The dual-instrument mean is used in the denominator as ACE-FTS and most other INST retrievals allow for negative concentrations. When the average of two compared values is near zero (due to allowing negative values), this can cause unrealistically large percent differences. When determining the multi-instrument averages of the ACE-FTS comparison results, the INST values (relative to ACE-FTS)

are weighted using the inverse square of the standard deviation of the differences,

$$W(z)_{INST}^{comp} = \frac{1}{\sigma(z)_{INST}^2}, \tag{2}$$





where $W(z)_{INST}^{comp}$ is the INST weight at height $z$, and $\sigma(z)$ is the standard deviation of the differences between INST and ACE-FTS.

Similar to the analyses by Hubert et al. (2016), the drift and its corresponding error at each altitude are determined by fitting the 30-day mean relative difference time series to a linear model using iterative reweighting least-squares fitting with a bisquare

weighting function (Street et al., 1988). Using 30-day mean values as opposed to daily means eliminates the need to deseasonalize the data prior to analysis. A Student's t-test is then calculated to determine the uncertainty of the linear drift for a confidence level of 99%. When determining the multi-instrument average of the ACE-FTS drift, the drift values for ACE-FTS – INST are weighted using the inverse square of the uncertainty of the drift,

$$W(z)_{INST}^{drft} = \frac{1}{\delta(z)_{INST}^2},$$   (2)

where $W(z)_{INST}^{drft}$ is the INST weight at height $z$, and $\delta(z)_{INST}$ is the uncertainty of the calculated linear drift between INST and ACE-FTS. The drift is considered to be non-zero when the weighted-average error bounds are less than 100% of the drift value.

Prior to analysis, the ACE-FTS data was screened using the ACE-FTS quality flags as described by Sheese et al. (2015). For all other data sets, all recommended quality, status, and convergence flags were taken into account when such flags were

available. Quality flags were generated for the MAESTRO data using the same algorithm as the ACE-FTS quality flags and were used to screen out unphysical outliers. For MLS data, no profile that was flagged as having cloud contamination at any pressure level was used in the analysis. For SABER data, as recommended by Fytterer et al. (2015), no data with values greater than 20 ppmv were used. Only SMR data where the corresponding measurement response (sum of the rows of the averaging kernels) values were greater than 0.8 were used.

**4 Results**

**4.1 Global comparisons**

Figure 1 shows the results of comparing v3.6 and v4.1 ACE-FTS $O_3$ profiles to coincident MAESTRO, MLS, OSIRIS, SABER, and SMR using all available data from 2004 to 2020 with coincidence criteria of within 6 hours and 300 km (excluding MAESTRO PM data below 23 km, as will be discussed in Sect. 4.2). The thick black lines are the multi-instrument weighted

averages and are shown without the individual INST comparisons in Fig 2. As can be seen in Fig. 1, both versions 3.6 and 4.1 yield similar average standard deviations of differences and correlation coefficient profiles. At all altitudes examined the correlation coefficients are on the order of 0.8-0.9, and the standard deviations are on the order of 15% (~0.3 ppmv) in the upper stratosphere, 10% (~0.5 ppmv) in the middle stratosphere, and 20% (0.1 ppmv) in the lower stratosphere. It should be noted that the coincidence criteria were chosen so that the estimated 1 σ natural variability (variability due to sampling

differences) is less than or on the order of ~5% (Sheese et al., 2021). Figure 2 highlights the difference in the ACE-FTS bias between versions 3.6 and 4.1, showing that the bias in the upper and lower stratosphere improved in 4.1, but worsened in the





middle stratosphere. In the 15-20 km region, the bias decreased from ~3-5% (~0.05 ppmv) to about -1 to 5% (-0.02 to 0.05 ppmv) and is in part due to updates in spectroscopic parameters from HITRAN2004 to HITRAN2016. Above 45 km the bias decreased slightly from 2-13% to 2-11% due to an improvement in the ACE-FTS altitude registration (Boone et al., 2020). In the 20-45 km region, where ozone concentrations peak, the bias was on the order of -1 to 3% (maximum of 0.2 ppmv near 33

km) in v3.6 and increased to ~2-9% (maximum of just over 0.5 ppmv near 30 km). The increase is due to the improved instrument line shape modelling. Preliminary uncertainty budget analysis suggests that errors in the field-of-view (FOV) modelling in the ACE-FTS $O_3$ retrievals can lead to a positive bias that peaks at 7% near 27 km and decreases to ~2% at 20 and 40 km. The FOV modelling in both the v3.6 and v4.1 algorithms, use only one ray through the tangent height to model the ACE-FTS observations instead of using multiple rays, which improves the vertical sampling. A study on this topic is

currently in preparation. Figure 3 shows the weighted averages of the ACE-FTS biases corrected for FOV modelling bias. With the correction, the bias gets worse in the upper stratosphere, increasing to 16 and 15% for v3.6 and v4.1 respectively at 53 km, and improves the v4.1 bias in the middle stratosphere where it is typically less than 5% between 16 and 48 km.

Figure 4 shows the calculated v3.6 and v4.1 ACE-FTS drift profiles relative to INST, given the coincidence criteria of within 6 h and 300 km. All INST profiles exhibit similar differences in drifts versus ACE-FTS profiles between v3.6 and v4.1. All

ACE-FTS – INST profiles show no significant change in drift between ACE-FTS data versions below ~22 km and a significant positive shift in drift, on the order of 1-3% dec$^{-1}$, above ~22 km. These changes in drift between ACE-FTS versions are more clearly seen in Fig. 5, which shows that the largest differences in drift are exhibited in the 40-45 km region. Also shown is the inter-instrumental stability, which is calculated (for both v3.6 and v4.1) as the standard deviation of all ACE-FTS drifts relative to INST. For both versions of ACE-FTS, the best inter-instrumental stability is 0.8% dec$^{-1}$ near 22 km, and between 17 and 46

km, the inter-instrumental stability is below 2% dec$^{-1}$. Figure 6 shows the ACE-FTS – MLS time series (30-day mean values) at 42.5 km as an example of the calculated trends and their uncertainty (represented as 99% confidence intervals). At this altitude level, the ACE-FTS v3.6 drift relative to MLS is -1.3±1.1% dec$^{-1}$ (-0.06±0.05 ppmv dec$^{-1}$), whereas the drift in v4.1 relative to MLS is not significant with a value of 0.9±1.3% dec$^{-1}$ (0.04±0.06 ppmv dec$^{-1}$). The global drift results, seen in Fig. 7, clearly show that not only is there a significant difference in drift values between v3.6 and v4.1 at nearly all altitudes above

~25 km, but that there is a significant drift in v3.6 $O_3$ at all altitudes above 20 km and no drift in v4.1 at any analysed altitude. ACE-FTS v3.6 data exhibits a mean drift on the order of 1±0.5% dec$^{-1}$ between 20 and 35 km and on the order of 2.5±1% dec$^{-1}$ in the 40-50 km region. This drift is due to an inaccurate trend in the assumed $CO_2$ concentrations used in the v3.6 pressure/temperature retrieval (which are used in the $O_3$ retrievals). In version v4.1, $CO_2$ concentrations are determined using a more accurate model (Boone et al., 2020).

**4.2 Regional comparisons**

The ACE-FTS weighted-average bias was calculated for data that were binned by ACE-FTS measurement local time (AM and PM) and by latitude—Arctic (50-90°N; Arc), Antarctic (50-90°S; Ant), and extra-polar (50°S-50°N; EP). The results, Fig. 8, show that in the middle stratosphere there is little change with local time and latitude in the ACE-FTS bias. At all local times



and latitudes between 20-40 km, the v3.6 bias is typically positive and on the order of 0-4%, and the v4.1 bias is positive and greater than 1%, peaking near 30 km at ~8-9%. As shown in Sect. 4.1, these biases include a positive bias introduced by FOV modelling error, which, for global averaged data, is on the order of a couple percent near 20 and 40 km and ~7% near 27 km. The weighted-average biases seen in Fig. 8 have not been corrected for FOV modelling errors, as those errors have not yet

been calculated on a regional basis.

When comparing ACE-FTS $O_3$ to INST profiles that were binned according to measurement local time, MAESTRO was the only instrument that exhibited a significant difference in drift values with respect to ACE-FTS between AM and PM measurements. As seen in Fig. 9, below 23 km, the PM comparisons exhibit a statistically significant drift on the order of 5-10%, whereas the AM comparisons exhibit a non-significant drift of ~1-2%. As none of the other instruments exhibit this

type of difference with respect to local time (not shown), it is likely that it is the MAESTRO PM that data have a positive drift. Due to this, all global comparisons in Sect. 4.1 exclude MAESTRO PM data below 23 km. The source of this drift has not yet been identified but is being investigated by the MAESTRO team.

The weighted average ACE-FTS drift was calculated for different local times and latitude regions for v4.1 only. The v3.6 drift identified in Sect. 4.1 exists at all local time and latitudes and therefore the v3.6 $O_3$ data are not recommended for use in trend

studies. As seen in Fig. 10, when the v4.1 data are binned by local time there are small but significant drifts of approximately -0.8±0.8% dec⁻¹ in the AM data near 21 km and approximately -1±1% dec⁻¹ in the PM data at 44.5 km. When the AM and PM data were further partitioned into Arctic, Antarctic, and extra-polar latitudinal bins, no significant trend was detectable near 45 km in the PM data at any latitude region, and the AM data only exhibited a significant drift in the extra-polar region near 21 km, as shown in Fig. 11. Unfortunately, the ACE sampling is too sparse to detect a statistically significant

drift result in sub-regions of the AM EP bin, which could help to further elucidate the cause of the drift. However, it should be noted that in every latitudinal region there are no significant differences between AM and PM mean drifts, and similarly, for both AM and PM data there are no significant differences in drift between latitudinal regions.

The Global Climate Observing System (GCOS) recommends that for long term stratospheric $O_3$ trend studies, data sets should have a stability of better than 1% dec⁻¹ (GCOS, 2011), and the European Space Agency Ozone Climate Change Initiative

program (ozone_CCI) recommends a stability of less than 1-3% dec⁻¹ at all latitudes throughout the stratosphere (van Weele et al., 2016). Both of the significant AM and PM drifts detected in the ACE-FTS v4.1 $O_3$ data (-1% dec⁻¹ near 45 km and 0.8% dec⁻¹ near 21 km, respectively) are at or below the recommended limits.

Figure 11 also shows that in the upper altitude regions, above 50 km, the Antarctic data exhibit a positive drift on the order of 2-4% dec⁻¹, which is more prominent for PM data. When the data are not binned by local time, the SH results exhibit a positive

drift that is significant above 51 km and has a maximum value of 2.4±1.5%, which is within the ozone_CCI stability recommendations but not the more strict recommendations of GCOS.



## 5 Summary

Both versions 3.6 and 4.1 of stratospheric ACE-FTS $O_3$ profiles have been compared to correlative measurements from the limb-viewing satellite instruments MAESTRO, MLS, OSIRIS, SABER, and SMR. On a global scale, the v3.6 profiles exhibit a mean bias that is positive on the order of ~3% in the 17-35 km region and a bias that increases with altitude from -1% near

40 km up to ~13% near 52 km. When the bias profile is corrected for FOV modelling errors, the v3.6 bias is ~5% below 22 km, within -4-0% between 23 and 43 km, and increases up to 16% at 53 km. The v4.1 $O_3$ bias profile tends to be more positive than that of v3.6 in the middle stratosphere, reaching up to 9% near 30 km, and more negative by ~3-4% percent above 46 km as well as below 20 km. When the v4.1 bias is corrected for FOV modelling errors, it is typically positive throughout the stratosphere and within 0-5% between 16 and 48 km and increases to ~15% near 53 km.

In the middle stratosphere (~20-45 km), neither version varies drastically (typically <3%) as a function of local time or latitude, however above 45 km and below 20 km, there are significant differences in the mean biases between AM and PM profiles. Below 20 km, the bias in the AM data is up to 5% greater than the PM bias, and above 45 km the PM bias is up to ~6% greater than the AM bias. The GCOS recommendations for $O_3$ profile accuracy are within 5-20% in the upper stratosphere and within 10% in the upper troposphere – lower stratosphere (GCOS, 2011). At all analysed altitude levels both the v3.6 and v4.1 $O_3$

data meet these accuracy recommendations, regardless of whether the data are corrected for FOV forward modelling errors or not. ESA ozone_CCI however has stricter recommendations. Above 20 km, ozone_CCI recommends an $O_3$ accuracy of within 8%, and an accuracy of 16% below 20 km. The v3.6 PM data meet these requirements at all analysed altitude levels below 47 km, and at all altitude levels below 51 km for the AM data. The v4.1 data, uncorrected for FOV modelling bias, however, do not meet the ozone_CCI requirements near 30 km in all sub-regions, as the bias can range between 7 and 9%, depending

on local time and latitude. The v4.1 data near 30 km only meet the ozone_CCI recommendations of accuracy after being corrected for FOV modelling bias.  However, it should be noted that the results presented in this study are not strictly measurements of accuracy, rather of bias relative to other limb sounders where the true $O_3$ profiles are unknown.

In terms of drift, ACE-FTS v4.1 $O_3$ is a significant improvement over v3.6. The v3.6 data exhibited a significant drift at all analysed altitude levels above 21 km, peaking at -0.14±0.02 ppmv dec$^{-1}$ at 40.5 km, and on the order of -1.0±0.4% within

~20-35 km and -2.5±0.8% dec$^{-1}$ within ~40-50 km. The ~-2.5% dec$^{-1}$ drift in the upper stratosphere technically meets the ozone_CCI stability recommendation of within 1-3% dec$^{-1}$ but not the GCOS recommendation of within 1% dec$^{-1}$. Therefore, v3.6 ACE-FTS $O_3$ above 35 km are not recommended for use in trend studies and should only be used with caution in the 20-35 km region. The v4.1 data, when analysed on a global scale exhibit no significant drift at any analysed altitude and drift values are within ±0.4% below 52 km. When the data are binned by local time, there is a small drift in the AM data near 21

km of approximately -0.8±0.8% dec$^{-1}$ and approximately -1±1% dec$^{-1}$ near 44 km in the PM data, however both these subsets adhere to the GCOS and ozone_CCI stability recommendations. This study also found that there is likely a significant positive $O_3$ drift (~2-10% with respect to ACE-FTS) in the MAESTRO PM data below 23 km that worsens with decreasing altitude.



**Data availability**

The ACE-FTS and MAESTRO Level 2 data can be obtained via the ACE website (registration required): http://www.ace.uwaterloo.ca. The OSIRIS data can be obtained via http://odin-osiris.usask.ca (registration required). The Aura MLS data can be obtained from https://disc.gsfc.nasa.gov/. The SABER data can be downloaded from http://saber.gats-inc.com/data.php. The SMR data can be downloaded from https://odin.rss.chalmers.se/level2.

**Author contributions**

The study was designed by PES, and KAW. PES wrote the paper. PES performed the analyses. Satellite data used in this study were provided by CDB, AEB, DAD, LF, CTM, DM, JMR and JZ. Valuable comments on the paper were provided by KAW, CDB, AEB, DAD, LF, CTM, DM, JMR and JZ.

**Competing interests**

The authors declare that they have no conflict of interest.

**Acknowledgements**

This project was funded by the Canadian Space Agency (CSA). The Atmospheric Chemistry Experiment is a Canadian-led mission mainly supported by the CSA. We thank Peter Bernath, who is the PI of the ACE mission. Odin is a Swedish-led satellite project funded jointly by Sweden (Swedish National Space Board), Canada (CSA), France (Centre National d'Études Spatiales), and Finland (Tekes), with support by the 3rd party mission programme of the European Space Agency (ESA). Work at the Jet Propulsion Laboratory was performed under contract with the National Aeronautics and Space Administration.

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



**Tables**

| Instrument | Satellite | Observation method | $O_3$ version | Comparison coverage | Vertical resolution (km) | Max # of coincidences |
|---|---|---|---|---|---|---|
| ACE-FTS | SCISAT | Solar occultation | 3.6 and 4.1 | 85°S-87°N | 3-6 | - |
| MAESTRO | SCISAT | Solar occultation | 3.13 | 85°S-87°N | 1 | 55,254 |
| MLS | Aura | Limb emission | 5.1 | 82°S-82°N | 2.5-4 | 26,408 |
| OSIRIS | Odin | Limb scatter | 5.10 | 83°S-82°N | 2 | 4,329 |
| SABER | TIMED | Limb emission | 2.0 | 82°S-82°N | 2 | 19,079 |
| SMR | Odin | Limb emission | 3.0 | 83°S-82°N | 2-3 | 8,672 |

**Table 1: Selected details of the instruments and the O₃ data sets used in the comparisons.**



# Figures

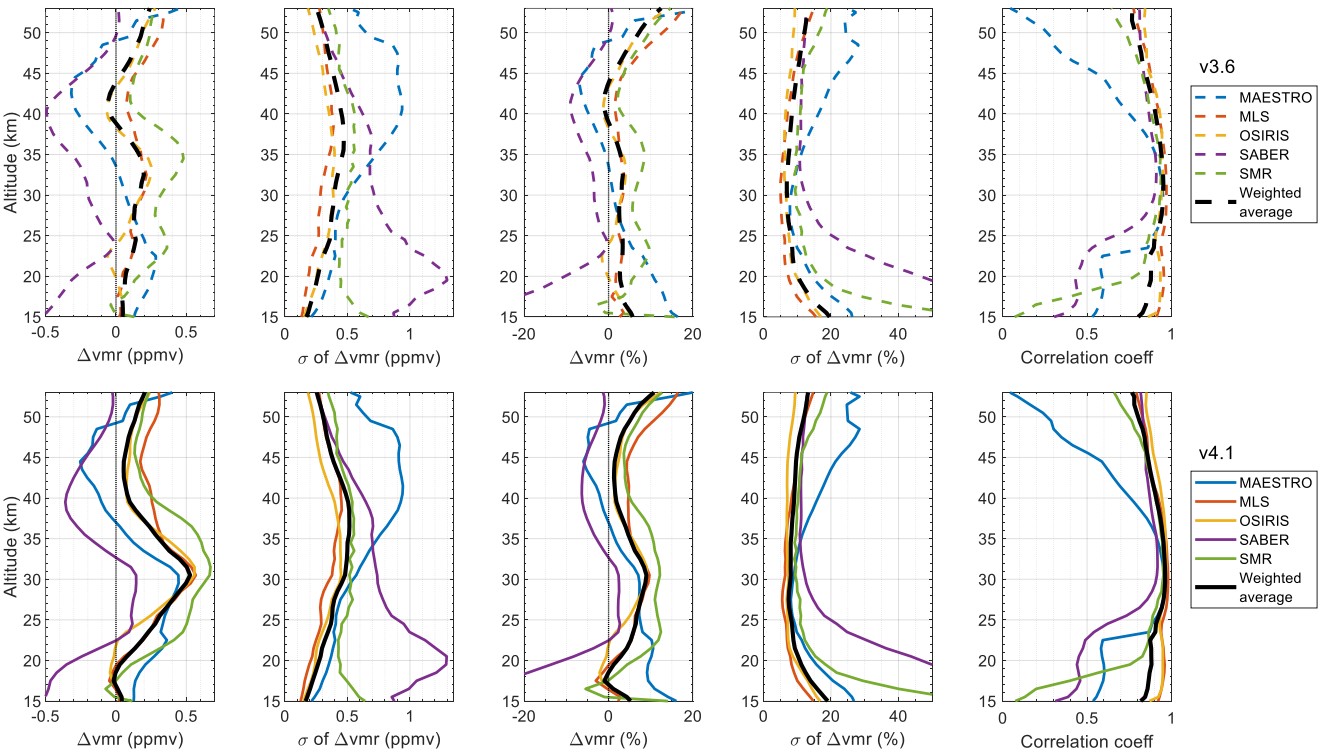

**Figure 1: Comparisons between ACE-FTS O₃ and MAESTRO (blue), MLS (red), OSIRIS (yellow), SABER (purple), and SMR (green) with coincidence criteria of within 6 h and 300 km for all coincident profiles within 2004-2020. (top) ACE-FTS v3.6 and (bottom) v4.1. From left to right the plots show the mean of the differences between ACE-FTS and INST in ppmv, the standard deviation of the differences between ACE-FTS and INST in ppmv, the mean of the relative differences between ACE-FTS and INST in percent, the standard deviation of the relative differences between ACE-FTS and INST in percent, and the correlation coefficients. The black line in each plot provides the weighted average (see text).**

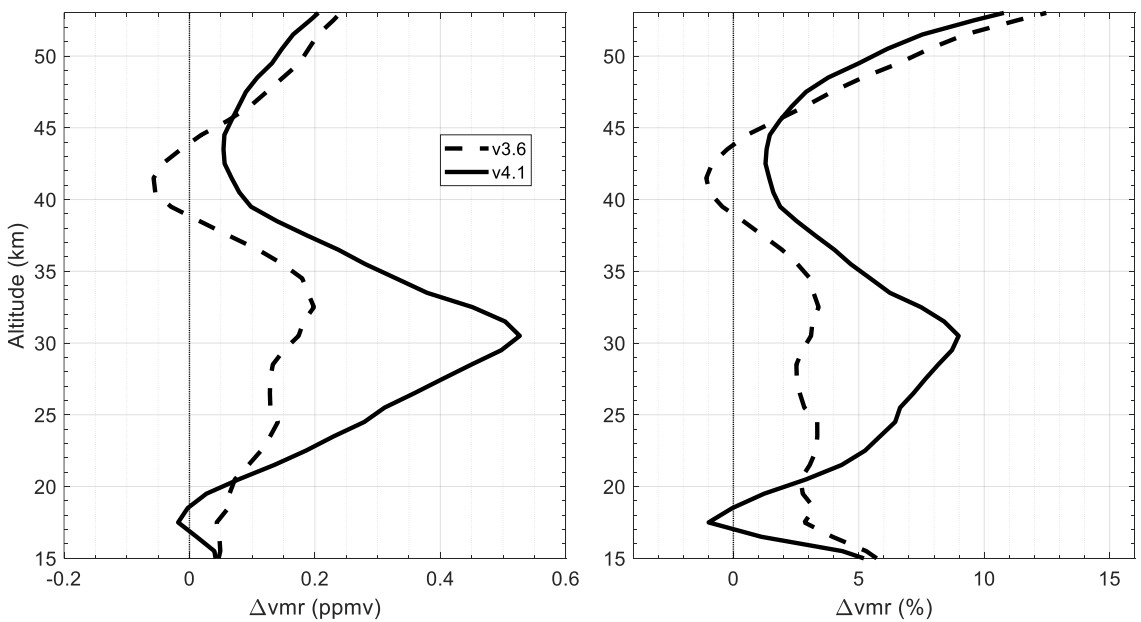

**Figure 2: Weighted averages of the mean differences (left) and mean percent differences (right) for comparisons between ACE-FTS and all instruments. Dashed lines represent ACE-FTS v3.6 and solid lines represent v4.1.**

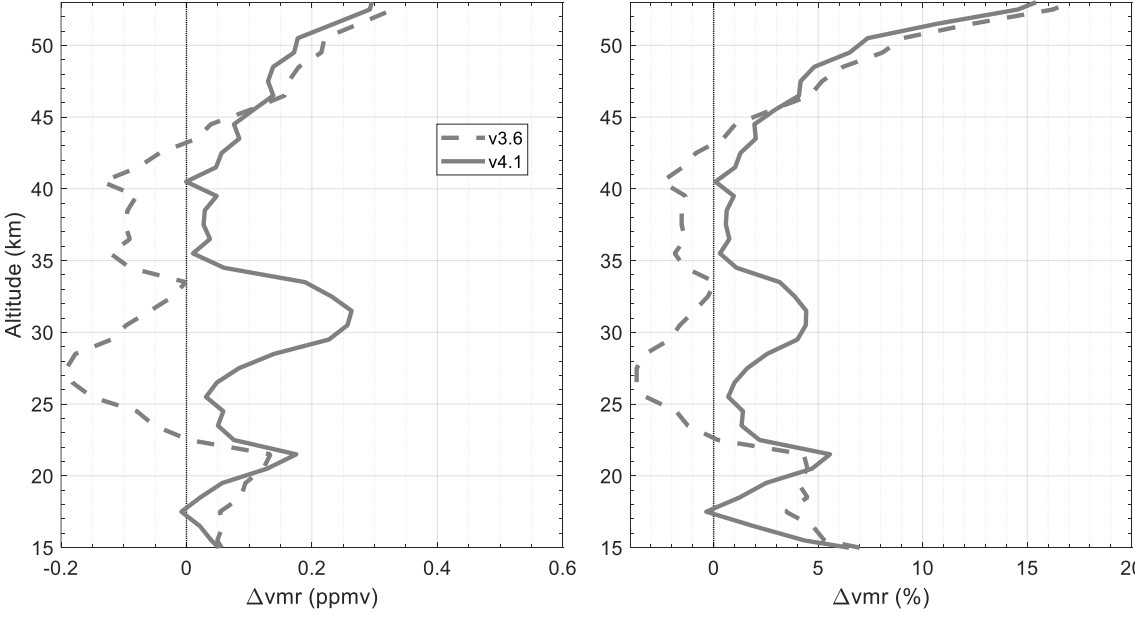

5   **Figure 3: Weighted averages of the mean differences (left) and mean percent differences (right) for comparisons between ACE-FTS and all instruments, corrected for FOV modelling bias. Dashed lines represent ACE-FTS v3.6 and solid lines represent v4.1.**





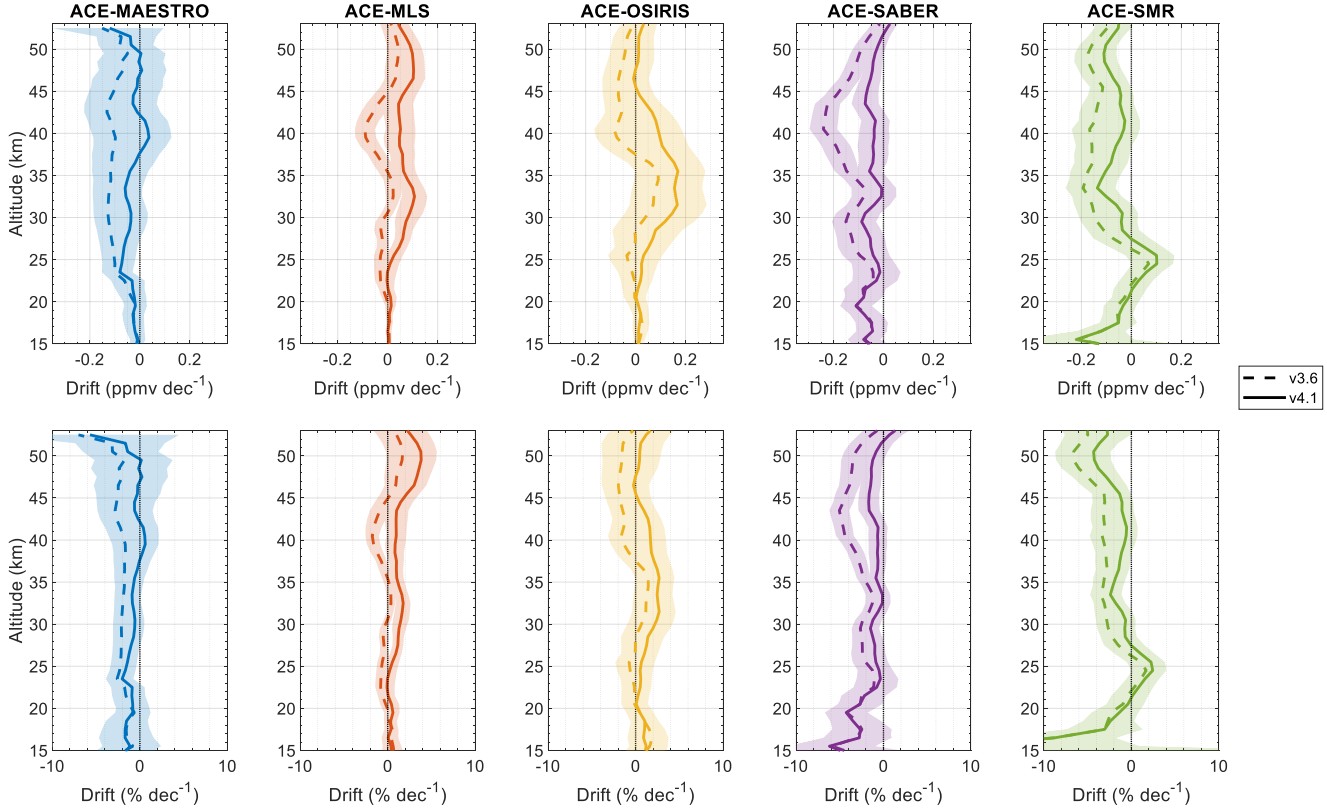

**Figure 4: Drift profiles (curves) and corresponding 99% confidence bounds (shaded regions) for comparisons between v3.6 and v4.1 ACE-FTS O₃ and MAESTRO (blue), MLS (red), OSIRIS (yellow), SABER (purple), and SMR (green) with coincidence criteria of within 6 h and 300 km for all coincident profiles within 2004-2020. (top) Results in ppmv dec⁻¹ and (bottom) in % dec⁻¹.**





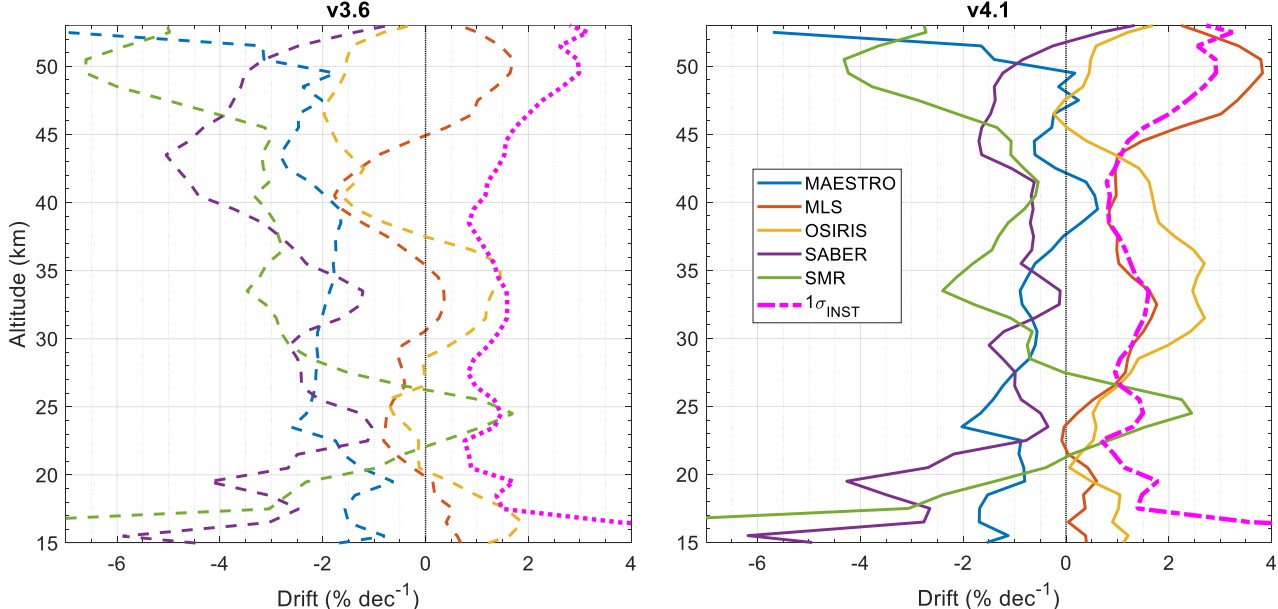

**Figure 5:** Relative drift profiles for each INST relative to ACE-FTS v3.6 (left) and v4.1 (right) and the corresponding inter-instrument stability (magenta), represented by the standard deviation of the drift profiles.



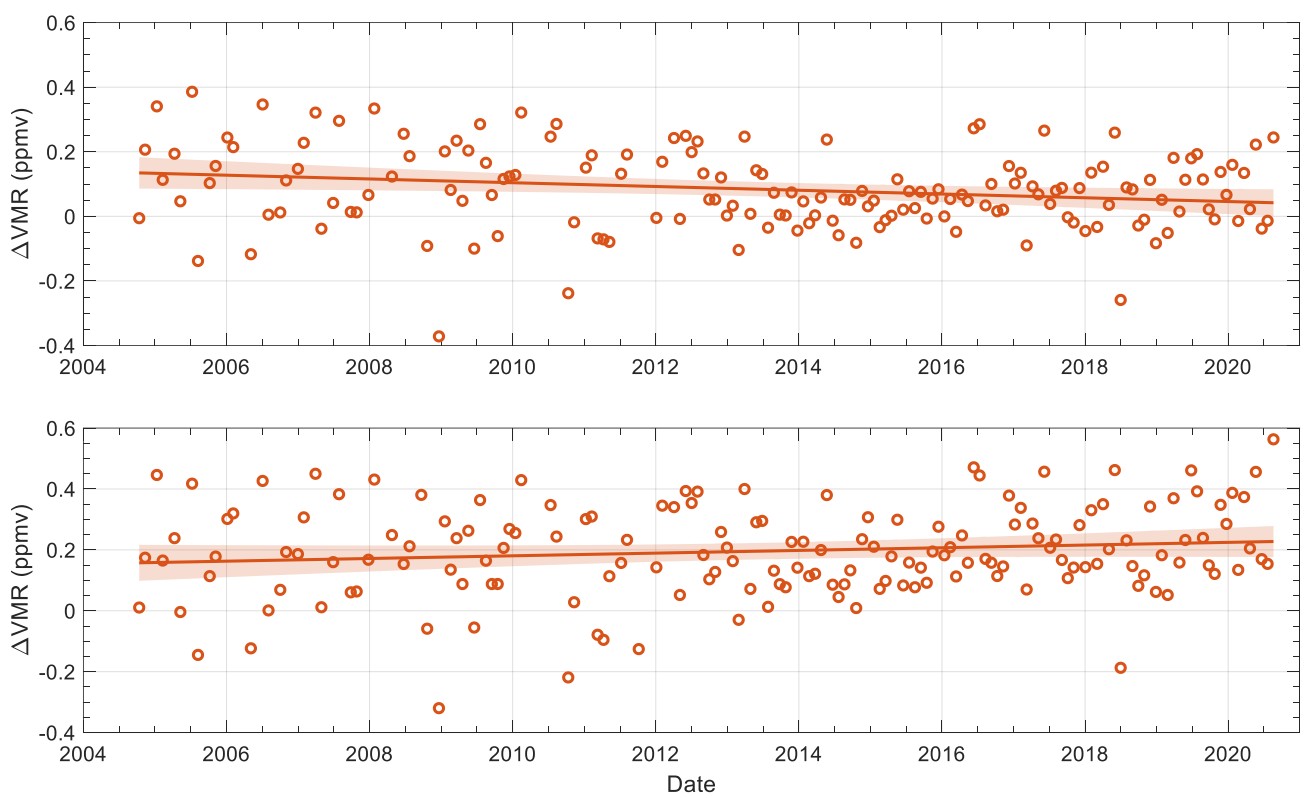

**Figure 6: Linear fits (lines) to the 30-day average differences (circles) between ACE-FTS and MLS at an altitude of 42.5 km for (top) ACE-FTS v3.6 and (bottom) ACE-FTS v4.1. Shaded areas represent the 99% confidence bounds.**

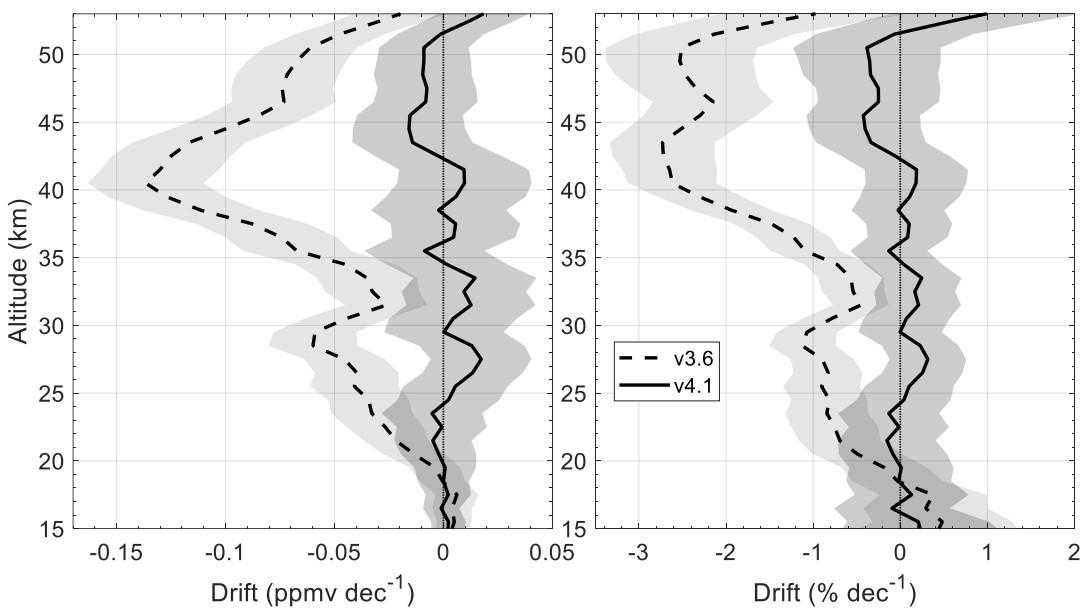

**Figure 7: Weighted average ACE-FTS drift profiles for v3.6 (dashed lines) and v4.1 (solid lines) in (left) absolute differences and (right) relative differences. Shaded regions represent the 99% confidence bounds.**

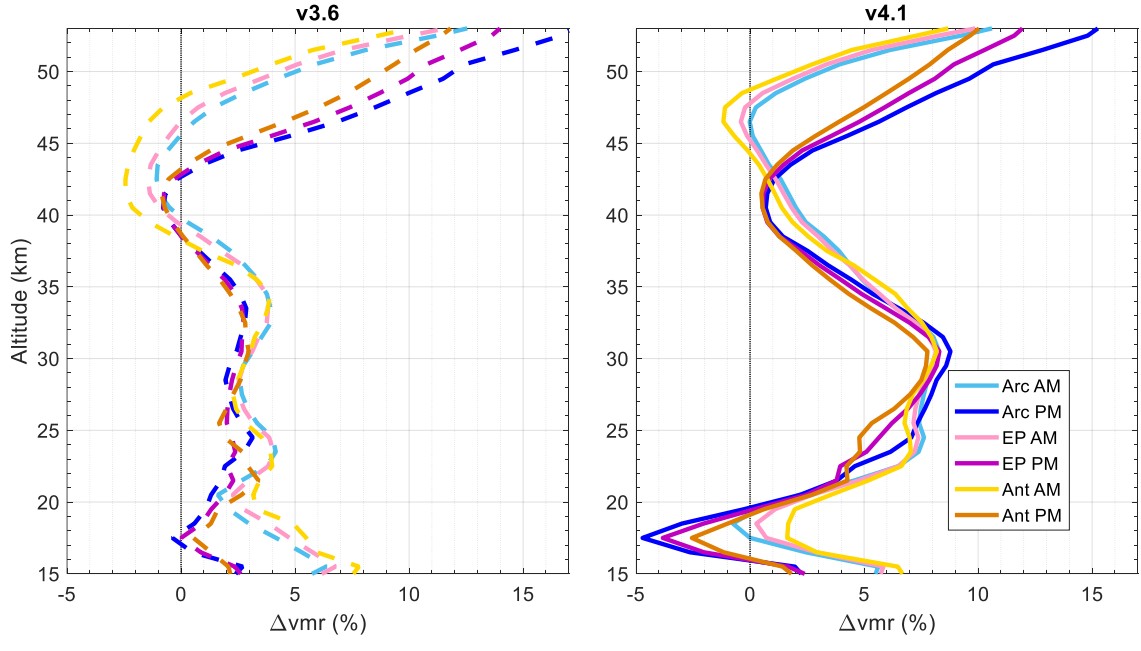

**Figure 8: Weighted-average percent biases between ACE-FTS O₃ and INST for comparisons in different local time and latitudinal bins of Arctic (Arc), extra-polar (EP), and Antarctic (Ant), and with coincidence criteria of within 6 h and 300 km. (left) ACE-FTS v3.6 and (right) v4.1.**



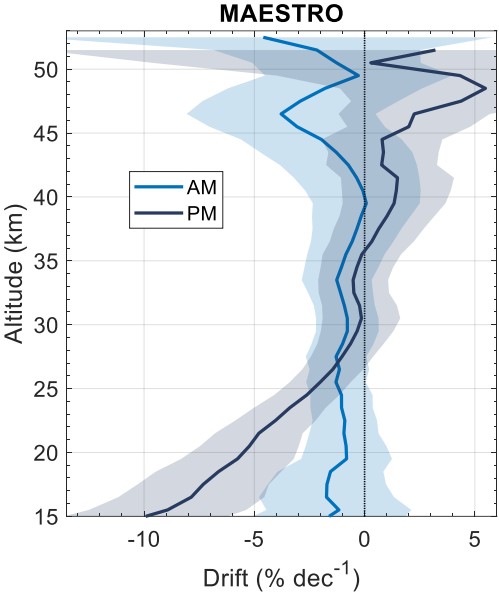

**Figure 9: Drift profiles between ACE-FTS v4.1 and MAESTRO for AM (blue) and PM (purple) measurements. Shaded regions represent 99% confidence bounds.**

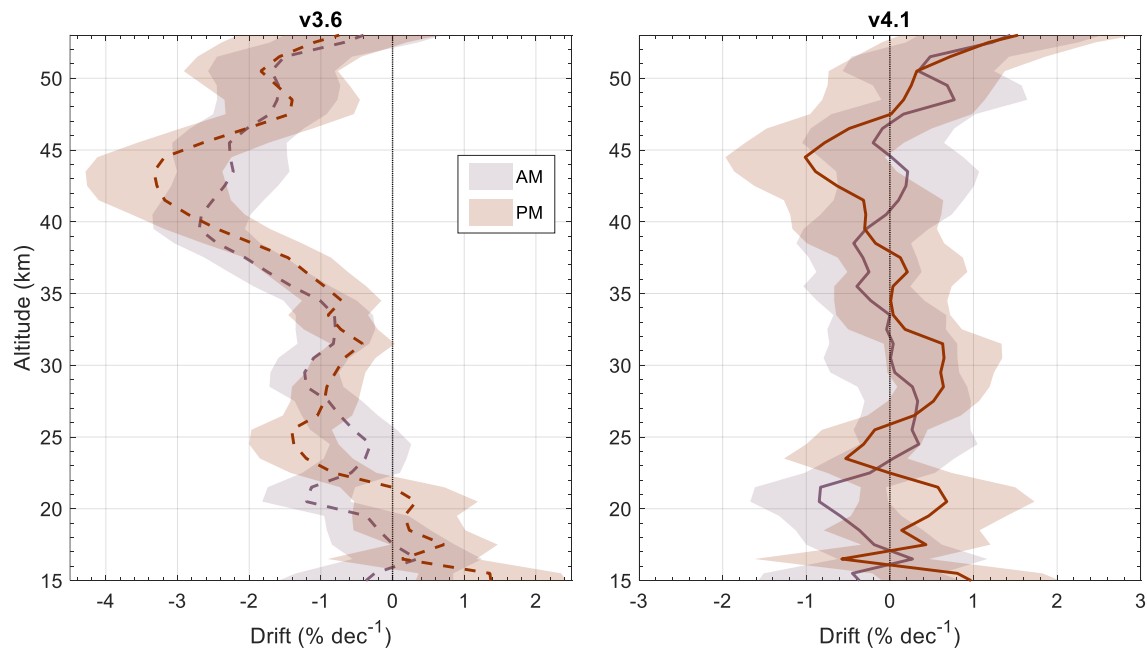

5 **Figure 10: Weighted-average ACE-FTS drift profiles for AM and PM comparisons in relative terms. (left) ACE-FTS v3.6 (right) v4.1. Shaded regions represent 99% confidence bounds.**



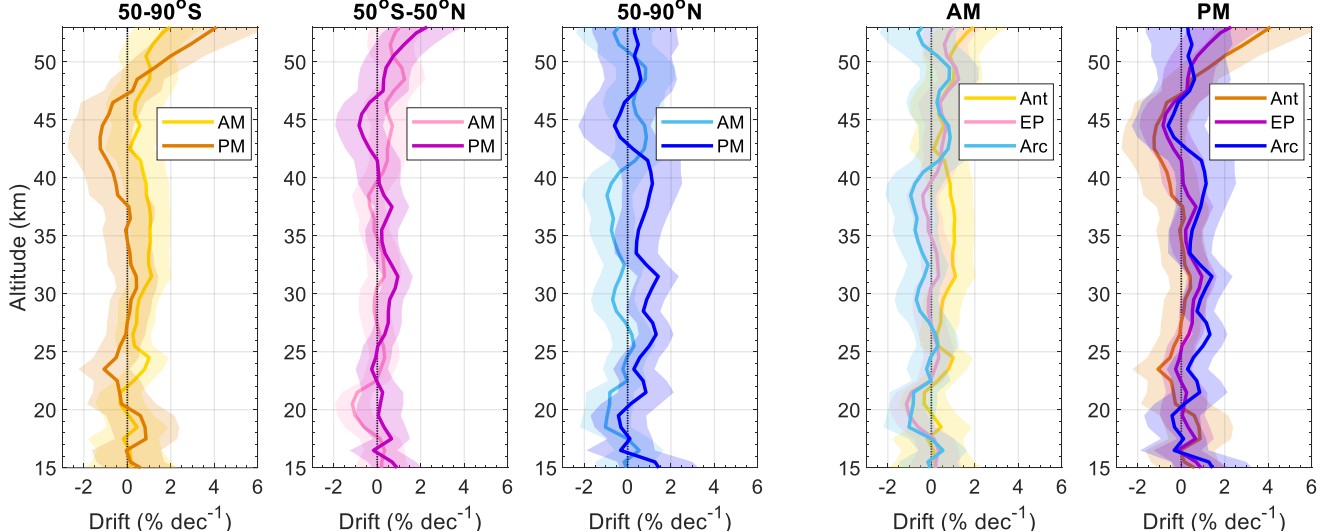

**Figure 11: Weighted-average ACE-FTS v4.1 drift profiles for comparisons in Antarctic (Ant), extra-polar (EP), and Arctic (Arc) latitude bands and separated by AM and PM local times. Shaded regions represent 99% confidence bounds.**