# Peer review of "Assessment of the quality of ACE-FTS stratospheric ozone data"

_Atmospheric Measurement Techniques, 2021_

## Referee Comment (RC1)

Review of Sheese et al. (submitted to AMT)

This is an important validation paper and the authors have done a nice job of putting the results into context using standards set by GCOS, for example.

P1L19: Is "dec" accepted as an abbreviation of decade? The editor can confirm.

P1L23: "being corrected for field-of-view modelling errors" -> "accounting for the finite height of the field-of-view"          (my objection is that "errors" is plural)

P3L3: Delete "Since February 2004,"                (repetitive)

P3L5: "orbital geometry" -> "orbital geometry and tangent height"        (a suggestion since 6 km vertical sampling is very rare in the lower stratosphere for ACE-FTS)

P3L18: "accounted for" -> "taken into account"

P3L25: "210-1025" -> "285-1015"                (see 1$^{st}$ page of McElroy et al., 2007)

P4L13: use "observations" or "measurements" in this sentence, not both.

P4L30: I think the upper tangent height for OSIRIS is ~70 km in the default "strat" mode, although there are many scans that go to ~100 km in the less used "strat-meso" mode. Has that changed in the later years?

P5L4: "and are retrieved simultaneously" is in disagreement with Degenstein et al. (2009), which states that aerosol and $NO_2$ are "pre-retrieved".

P5L13: tuneable -> tunable

P5L29 (and elsewhere): Insert a space between the numeric value and the unit (μm)

P6L4: Don't capitalize "Personal"

P6L12: No need for a hyphen

P6L29: "of the ACE-FTS comparison results" -> "for the ACE-FTS comparison"

P7L5: "Using 30-day mean…" Why is there a need to deseasonalize the data when daily mean values are used? Spatially, is the drift calculated using relative differences over the globe, or relative differences within a latitude band (i.e., zonally). It becomes clear in section 4.1 and 4.2, but at this point, the reader might be wondering already. If data from both hemispheres are used (i.e. global data), then seasonality largely cancels, does it not?

P7L7: Why is 99% confidence chosen? Does this follow Hubert et al. or some other previous work? If so, please state "following Hubert et al. (2016)", for example.  I think I would be more interested in 95% confidence intervals.

P7L26: examined -> examined,

P8L5: This sentence is a bit odd (revision is suggested but not necessary). If the line shape is improved, then it comes as a surprise that the bias is larger.

P8L9: "which improves the vertical sampling" is not correct. The vertical sampling relates to the measurement, not the modelling.

P8L13: Why not indent at the start of a paragraph? I believe this is conventional for AMT.

P9L12: The authors have provided important information on a drift with MAESTRO PM measurements.

P9L30: (2.4±1.5)% is not different from 1%.

P10L18: I think it is worth mention that ozone has a significant diurnal variation above ~42 km and that some differences with other instruments may be due to the temporal coincidence criterion of 6 hours. Also, ACE-FTS may have a retrieval issue due to horizontal inhomogeneity across the day-night terminator at these altitudes (and higher).

P22 (Fig. 4): The confidence bounds are the same colour for v3.6 and v4.1. At some altitudes for some correlative sensors, the confidence bounds are overlapping for the two ACE-FTS versions. I recommend using an outline on the bounds for v4.1 so that they can be distinguished from the v3.6 confidence interval profile.

P26 (Figs. 9-10): Please explain the third shading colour in the caption…. maybe something like "overlapping confidence intervals"

P27: Although no one would consider 50-60°N to be Arctic and 50-60°S to be Antarctic, I think it is acceptable here.

---

## Author Comment (AC1)

**Response to Reviewers – "Assessment of the quality of ACE-FTS stratospheric ozone data" by Patrick E. Sheese et al.**

*We'd like to thank the reviewers for their helpful comments. Here we address the main and specific comments of each reviewer, with their comments in black and our responses in green.*

There has been one significant update since submitting this paper. The FOV modelling error calculations have been updated and the effect is significantly less than previously calculated. Therefore, we are no longer including this contribution in the analysis.

**Reviewer 1**

P1L19: Is "dec" accepted as an abbreviation of decade? The editor can confirm.
We'll keep "dec" (all instances) for now and can change later if the editor has an issue with it.

P1L23: "being corrected for field-of-view modelling errors" -> "accounting for the finite height of the field-of-view" (my objection is that "errors" is plural)
This is no longer discussed in the text, as noted above.

P3L3: Delete "Since February 2004," (repetitive)
This has been deleted.

P3L5: "orbital geometry" -> "orbital geometry and tangent height" (a suggestion since 6 km vertical sampling is very rare in the lower stratosphere for ACE-FTS)
This has been added.

P3L18: "accounted for" -> "taken into account"
This has been changed.

P3L25: "210-1025" -> "285-1015" (see 1st page of McElroy et al., 2007)
This has been corrected.

P4L13: use "observations" or "measurements" in this sentence, not both.
"observations from" has been deleted.

P4L30: I think the upper tangent height for OSIRIS is ~70 km in the default "strat" mode, although there are many scans that go to ~100 km in the less used "strat-meso" mode. Has that changed in the later years?
OSIRIS certainly scans higher than 60 km (as mentioned in the previous sentence), but the standard ozone retrievals only go up to 60 km.

P5L4: "and are retrieved simultaneously" is in disagreement with Degenstein et al. (2009), which states that aerosol and $NO_2$ are "pre-retrieved".
This has been corrected.

P5L13: tuneable -> tunable
This has not been changed as "tunable" is the American spelling, which AMT doesn't use.

P5L29 (and elsewhere): Insert a space between the numeric value and the unit (µm)
This, and all other instances, has been corrected.

P6L4: Don't capitalize "Personal"
Corrected.

P6L12: No need for a hyphen
Corrected.

P6L29: "of the ACE-FTS comparison results" -> "for the ACE-FTS comparison"
Changed.

P7L5: "Using 30-day mean…" Why is there a need to deseasonalize the data when daily mean values are used? Spatially, is the drift calculated using relative differences over the globe, or relative differences within a latitude band (i.e., zonally). It becomes clear in section 4.1 and 4.2, but at this point, the reader might be wondering already. If data from both hemispheres are used (i.e. global data), then seasonality largely cancels, does it not?
It could potentially be necessary if the "global" analysis is biased to a certain latitude region. Either way, this now reads "eliminates the need to deseasonalize the data prior to analysis within specific latitude bands."

P7L7: Why is 99% confidence chosen? Does this follow Hubert et al. or some other previous work? If so, please state "following Hubert et al. (2016)", for example. I think I would be more interested in 95% confidence intervals.
Hubert et al. went with 95%. We wanted to use an abundance of caution, just to be completely sure we're not reporting any drifts that might not actually exist. 99% confidence gives us that slightly more room, but it doesn't dramatically change any of the results from if we used 95%. The significant result here (regardless of the confidence interval chosen) is that v3.6 had a clear significant drift, which was ameliorated in v4.1. If necessary, we can redo the drift sections using 95%, but we'd prefer to keep it as is.

P7L26: examined -> examined,
Corrected

P8L5: This sentence is a bit odd (revision is suggested but not necessary). If the line shape is improved, then it comes as a surprise that the bias is larger.

The line shape was improved. The increase in bias relative to the other sounders indicates that there are likely still other biases not accounted for. This now states, "Although the overall bias with respect to other limb sounders worsened, the increase is in part due to…"

P8L9: "which improves the vertical sampling" is not correct. The vertical sampling relates to the measurement, not the modelling.

This is no longer discussed in the text, as noted above.

P8L13: Why not indent at the start of a paragraph? I believe this is conventional for AMT.

We're using the AMT template, which does not use indents at the start of paragraphs.

P9L12: The authors have provided important information on a drift with MAESTRO PM measurements.

Thanks!

P9L30: (2.4±1.5)% is not different from 1%.

True. We now state, "only just meets the stricter recommendations of GCOS within the uncertainty."

P10L18: I think it is worth mention that ozone has a significant diurnal variation above ~42 km and that some differences with other instruments may be due to the temporal coincidence criterion of 6 hours. Also, ACE-FTS may have a retrieval issue due to horizontal inhomogeneity across the day-night terminator at these altitudes (and higher).

This section now states "It should be noted that above ~40 km $O_3$ undergoes large diurnal variation and the time coincidence criterion of 6 h could be a source of the larger bias at these altitudes. Similarly, ACE-FTS does not account for the increased horizontal inhomogeneity across the day-night terminator at these altitudes."

P22 (Fig. 4): The confidence bounds are the same colour for v3.6 and v4.1. At some altitudes for some correlative sensors, the confidence bounds are overlapping for the two ACE-FTS versions. I recommend using an outline on the bounds for v4.1 so that they can be distinguished from the v3.6 confidence interval profile.

This was an issue with exporting the figure. In MatLab, the overlapping areas were darker. We have used a different exporting method, and now the overlap is clearer.

P26 (Figs. 9-10): Please explain the third shading colour in the caption…. maybe something like "overlapping confidence intervals"

Captions for Figures 4, 7, and 9-11 now indicate that the shaded regions are semitransparent.

P27: Although no one would consider 50-60°N to be Arctic and 50-60°S to be Antarctic, I think it is acceptable here.

It is not clear what this comment is referring to, as the text and figures all define the regions as 50-90°.

**Reviewer 2**

General comments:

While I appreciate a short-and-to-the-point paper as much as anyone, I felt this paper could have used a little more elaboration in places. I have outlined some specific examples below, but there are likely other places that would benefit.
The specific issues have been addressed below, and we now also include ACE-FTS relative precision estimates.

The title indicates an assessment of quality, but quality also covers precision which really isn't discussed. With relatively little extra effort an approach along the lines of Bourassa et al. (2012) section 3.2 would have been a very nice addition.
Thank you for the suggestion. We have included relative precision estimates following this methodology, and it is now discussed throughout the paper.

In light of this, a better title would be "Assessment of the accuracy …" or even something like "Assessment of the absolutely quality …" Something to consider…
We now include precision estimates so the title has remained the same.

Specific comments:

Page 7, line 30: So was 5% chosen so that it would be smaller than the observed % difference, which would imply that the instrument difference is the dominant contributor (assuming they would be added in quadrature)? Please be more explicit about the motivation for using 5%.
The text now reads, "For comparisons between atmospheric measurements, it is desirable to have coincidence criteria that allow for the geophysical variability to be less than the combined accuracies, and the value of 5% is the minimum accuracy recommended by the Global Climate Observing System (GGOS) for stratospheric $O_3$ (GCOS, 2011)."

Page 2, line 13 "… likely due to using too short a time…" Does this imply a significant drift would have been detected had the time series been long enough.
This now reads, "Although, as this study will show, it is possible that significant drifts would have been identified had longer time series been analyzed."

Equation (2): comment briefly on why this definition was chosen
The text now includes "This, rather than a simple mean, is done to account for the quality of the INST data sets used in the comparisons, and assumes that all data sets, in comparison with ACE-FTS, exhibit similar geophysical variability."

Page 7, line 6: "… is then performed …"
Corrected.

Page 8, line 5/6: "The increase is due to the improved instrument line shape modelling." – is this in the Boone et al. (2020) paper? Please elaborate a little;
This now reads, "The increase is in part due to the improved instrument line shape modelling, which now allows for asymmetry and improved wavenumber variation (Boone et al., 2020)."

also line 3: add some more about "ACE-FTS altitude registration" in the ACE-FTS data description section.
This now reads, "…the ACE-FTS altitude registration, which is now generated from measurements of the $N_2$ continuum (Boone et al., 2020)."